# Diversity and Evolutionary Analysis of Venom Insulin Derived from Cone Snails

**DOI:** 10.3390/toxins16010034

**Published:** 2024-01-09

**Authors:** Qiqi Guo, Meiling Huang, Ming Li, Jiao Chen, Shuanghuai Cheng, Linlin Ma, Bingmiao Gao

**Affiliations:** 1Key Laboratory of Tropical Translational Medicine of Ministry of Education, Hainan Key Laboratory for Research and Development of Tropical Herbs, School of Pharmacy, Hainan Medical University, Haikou 571199, China; qiqiguo@hainmc.edu.cn (Q.G.); mgling213@163.com (M.H.); 2022liming@gmail.com (M.L.); hy0207109@hainmc.edu.cn (J.C.); shuanghuaicheng2018@hainmc.edu.cn (S.C.); 2Griffith Institute for Drug Discovery (GRIDD), School of Environment and Science, Griffith University, Nathan, Brisbane, QLD 4111, Australia

**Keywords:** cone snails, insulin, conoinsulin, diversity, phylogenetic analysis

## Abstract

Cone snails possess a diverse array of novel peptide toxins, which selectively target ion channels and receptors in the nervous and cardiovascular systems. These numerous novel peptide toxins are a valuable resource for future marine drug development. In this review, we compared and analyzed the sequence diversity, three-dimensional structural variations, and evolutionary aspects of venom insulin derived from different cone snail species. The comparative analysis reveals that there are significant variations in the sequences and three-dimensional structures of venom insulins from cone snails with different feeding habits. Notably, the venom insulin of some piscivorous cone snails exhibits a greater similarity to humans and zebrafish insulins. It is important to emphasize that these venom insulins play a crucial role in the predatory strategies of these cone snails. Furthermore, a phylogenetic tree was constructed to trace the lineage of venom insulin sequences, shedding light on the evolutionary interconnections among cone snails with diverse diets.

## 1. Introduction

Cone snails thrive in various tropical and subtropical aquatic ecosystems, exhibiting a distribution that spans the coastal areas surrounding the South China Sea, Africa, the Red Sea, India, Ceylon, Japan, the Paracel Islands, and even extends as far north as the Croatian coast [1]. The number of identified species has increased significantly, from 500 species in 2004 to more than 900 species today, and research on *Conus* is still developing rapidly [2,3]. Cone snails have distinctive shell patterns and they can be classified. According to their dietary preferences by which they can be categorized into different groups, such as piscivorous (feeding on fish), vermivorous (feeding on worms), and molluscivorous (feeding on mollusks), thereby highlighting the various feeding habits within this snail species [2,4,5]. Figure 1 showcases several representative species of *Conus* with different feeding habits, including piscivorous cone snails such as *Conus geographus*, *Conus tulipa*, *Conus kinoshitai*, and *Conus striatus*, vermivorous cone snails such as *Conus floridulus*, *Conus imperialis*, *Conus quercinus* and *Conus eburneus*, and molluscivorous cone snails such as *Conus textile*, *Conus marmoreus*, *Conus episcopatus* and *Conus victoriae*. 

Cone snails have evolved to produce intricate varieties of toxins, thus serving as a prominent focus of contemporary scientific research. They diversified across various species and eras, producing a toxic system that offers distinct advantages [6,7,8,9,10,11]. Conotoxins act on ion channels and receptors in the nervous systems, including sodium ion channels (Nav), potassium ion channels (Kv), and calcium ion channels (Cav). Moreover, they exhibit an affinity for hormone receptors, including oxytocin and antidiuretic hormone [12,13,14,15]. Each *Conus* species’ venom contains a distinct array of pharmacologically active peptides, predominantly specific to that particular species [3,16]. Cumulatively, the 900 species within the *Conus* genus produce a significant quantity of these peptides, potentially surpassing a remarkable total of 180,000 distinct variants [17,18,19,20,21,22]. Cone snails employ their venomous arsenal for prey capture and defense against competitors, exhibiting a wide range of hunting strategies that include predation upon worms, snails, and fishes [23,24]. Additionally, they harness their venoms as a means to deter potential predators [11]. 

Vertebrates such as snakes and invertebrates such as cone snails, spiders, and scorpions are all important sources of venoms [25,26,27,28]. Currently, pharmaceuticals derived from venoms exhibit a broad spectrum of applications encompassing analgesic, anti-inflammatory, and antihypertensive properties [29,30,31]. Clinically approved drugs derived from animal venom components include captopril, enalapril, ziconotide, eptifibatide, tirofiban, lepirudin, bivalirudin, batroxobin, apitox, cobratid, and exenatide [32,33]. Ziconotide, the groundbreaking pharmaceutical derived from conotoxins, has been utilized since 2004 to address chronic pain by selectively inhibiting voltage-gated calcium channels. It obtained FDA approval for the treatment of chronic severe pain in patients who display intolerance or inadequate response to systemic analgesics or intrathecal morphine [34,35].

Exenatide, derived from a peptide (exendin-4), found in the saliva of the Gila monster, is one of the most classical examples of venom toxin derivatives [32,36]. It effectively regulates glycemic levels by stimulating glucose-dependent insulin secretion [37], and has been developed for the treatment of diabetes [36,38]. In 2005, the FDA approved its utilization as an adjunct therapy, paired with metformin or sulfonylurea, in the management of type 2 diabetes mellitus. Apart from *Heloderma’s* exendin-4, a diverse array of bioactive substances derived from cone snails, sea anemones, scorpions, mosquitoes, bees, and spiders exhibit promising prospects in effectively regulating blood glucose levels [32,39,40]. Based on the findings of these investigations, Safavi-Hemami et al. [41] identified insulin-like peptides, namely Con-Ins G1 and Con-Ins G2, within the venom of *C. geographus*. Con-Ins-G1 was observed to possess predatory adaptations by effectively reducing blood glucose levels in prey organisms as a means of capturing and immobilizing them [41]. Therefore, the identification of conotoxin-like insulins (conoinsulins) sourced from *Conus* species holds paramount importance in advancing the field of marine drugs.

In this review, a comprehensive comparative analysis was conducted to investigate the sequence and structural characteristics of insulin derived from human, zebrafish, and cone snails. All available insulin sequences were retrieved from databases, followed by comparisons and subsequent analysis of their diversity. A rich array of conoinsulin sequences from *Conus* specimens exhibiting diverse feeding habits were collected, subsequently enabling the construction of a phylogenetic tree to unveil their intricate evolutionary relationships. The primary objective of this paper is to establish a solid foundation for the exploration of conoinsulin’s diversity and phylogenetic analysis, thus furnishing providing robust support for the advancement of novel marine drug development endeavors.

## 2. Conoinsulin as a Weapon for Predation

Cone snails are commonly found in coastal reefs, rocks, and sandy beaches [42]. The shells of the cone snails are visually striking, featuring a diverse range of colors and bearing a resemblance to their non-toxic *Conus* counterparts. Consequently, they are occasionally inadvertently picked up by humans, resulting in potentially life-threatening incidents. There have been several documented incidents of accidental *Conus* collections leading to human fatalities [43]. Such poisonings are attributed to various toxins contained in the cone snail. This particular occurrence of poisoning is predominantly ascribed to various toxins manufactured within the cone snails. *C. geographus*, a proficient piscivorous predator, further exemplifies one of the most dangerous species to humans. Its venom can cause approximately 50% of poisoning incidents, as it encompasses 50% of the overall number of fatalities in all poisoning incidents [43]. Furthermore, *C. geographus* tends to result in higher mortality rates in children compared to adults [43]. Regardless of the victim’s age, the poisoning incidents from larger snails are typically more prevalent than those from smaller snails. Additionally, humans have also suffered stings from other fish-feeding *Conus* species, leading to fatal injuries [43]. The venom produced by other *Conus* species that prey on gastropods also poses significant safety hazards to humans. However, most reported deaths have yet to be confirmed.

The unique predation methods of cone snails have attracted significant interest from many researchers [44,45]. Some cone snails employ a burying behavior, concealing their bodies within the sand while exposing their elongated proboscis. This adaptive strategy enables them to simultaneously acquire oxygen and observe the activities of neighboring organisms. When detecting the proximity of potential prey, the cone snail deploys its elongated siphon beak in the direction of the prey [23]. Subsequently, the cone snails employ a specialized radular tooth with a hollow, harpoon-like structure, to immobilize the fish. This tooth not only anchors firmly onto its prey, but also injects a potent venom [23,44,45], acting much like a flexible "hypodermic syringe" connected to a sac filled with toxins [46]. By means of muscular contraction, the cone snail expels its venom into the prey’s body within a fraction of a second [47]. The venom discharged by the *Conus* species possesses potent toxic properties, capable of inducing severe poisoning or even lethality in the affected organism. 

Fish rely on their inherent biological nervous system to regulate and coordinate their physical movements. When penetrated by the cone snail’s harpoon-like structure, the fish experiences a transient period of uncontrolled locomotion, lasting for a fraction of a second, followed by an immediate and complete paralysis [46]. The mixture of conotoxins contained in the venom swiftly targets the chemical receptors and ion channels responsible for modulating the fish’s neural transmissions, causing the receptor to remain persistently open due to the influx of the toxin. The conotoxin, administered by the cone snail, induces muscular spasms in the fish. These toxins target the synaptic connections between the fish’s nerves and muscles, impeding the muscle tissue’s ability to receive command signals. Gradually, as the intensity of spasms diminishes, the fish typically succumbs to complete paralysis in the majority of observed instances [46,48]. Lastly, the cone snail retracts its proboscis and proceeds to maneuver the incapacitated prey towards its oral cavity, thereby concluding the entirety of the predation process (Figure 2A).

Furthermore, an alternative predatory tactic employed by certain *Conus* species, referred to as *C. geographus*, involves the conspicuous expansion of their oral aperture, colloquially referred to as their "bloody mouths." Once opened, these individuals commence the emission of venomous substances into the surrounding aqueous environment, effectively paralyzing an entire assemblage of fish. These immobilized fish are subsequently engulfed by the cone snail in a single ingestion motion, facilitating the swift consumption of multiple prey items (Figure 2B). 

Within the venomous arsenal of *C. geographus*, a multitude of bioactive compounds are present, including insulin. This species uses specialized insulin in its venom to facilitate hunting [41]. Fast-acting neurotoxin and delivery systems are required for cone snails to use conoinsulin to capture prey [49]. After locking on to the prey, the insulin in the venom is rapidly propelled toward the bulbous base of the harpoon, where it is injected into the fish [50]. Relevant studies and data have shown a very fast delivery speed of cone neurotoxin [47]. Remarkably, *C. geographus* produces two distinct forms of insulin. One resembles molluscan insulin and is produced in the nerve ring and the esophagus, which play a role in the regulation of hemolymph glucose levels, memory and learning. Another type of insulin, similar to the prey’s insulin, known as "venom insulin", is exclusively expressed in proximity to the injection site within the venom duct, attaining a conspicuously high concentration, which is the main component of the venom used for predation [41,51].

The fish-hunting cone snail, *C. geographus*, was previously shown to use derived venom insulin Con-Ins G1 to capture prey [14]. In vivo, intraperitoneal injection of Con-Ins G1 into zebrafish lowers their blood glucose with the same potency as human insulin [41,52]. When applied to water, Con-Ins G1 reduces the overall locomotor activity of zebrafish larvae, demonstrating a significant decrease in the percentage of time spent swimming and movement frequency [41]. When tested on a mouse model of diabetes, Con-Ins G1 also lowered blood glucose with a 10-fold higher potency compared to human insulin [52]. Con-Ins G1 has been demonstrated to constitute a prominent constituent within the venom composition of this particular species. These insulin-like molecules contained therein exert a pronounced effect on the blood glucose levels of the prey organisms, resulting in a significant reduction that impairs their ability to evade capture or flee from predation [41,53]. The comprehensive literature review reveals that the crude venom of cone snails targets not only voltage-gated ion channels, such as Nav, Kv and Cav channels, but also acts on nAChR and insulin receptors (IRs) [Figure 2C]. Therefore, conoinsulins, like other conotoxins, exhibit robust biological activity and hold potential as drug candidates for the treatment of diabetes, infertility, and other diseases. 

## 3. Diversity Analysis of Conoinsulin

Insulin-like peptides, identified in cone snail venom, exhibit species-specific variations regarding expression. Certain species exclusively express a single type of conoinsulin, whereas others possess multiple conoinsulin variants. The precursors of these conoinsulins typically feature a conserved N-terminal signal sequence in their amino acid sequences. Among these sequences, some contained a pro-peptide region, while others did not; however, all these sequences exhibit at least one amino acid difference in the mature region [54]. Cone snails deploy venoms containing conoinsulin, which acts within seconds to immobilize nearby fish, facilitating easier capture and consumption [41,55]. This rapid action of conoinsulin, in stark contrast to human insulin, has intrigued scientists. In the ConoServer and UniProt databases, 38 different insulin sequences and one insulin-like peptide have been reported in the venom of 18 types of *Conus*, and alignment analysis was performed using MEGA 7.0.14. Con-Ins G1, Con-Ins G1b, Con-Ins G1c, Con-Ins G2, Con-Ins G2b, Con-Ins G3, and Con-Ins G3b were all sourced from *C. geographus*; Con-Ins T1A, Con-Ins T1B, Con-Ins T2, Con-Ins T3, and Con-Ins T4 were all discovered in *C. tulipa*; Con-Ins K1 and Con-Ins K2 were derived from *C. kinoshitai*; Con-Ins Q1 and Con-Ins Q1b were both sourced from *C. quercinus*; Con-Ins F1, Con-Ins F2, Con-Ins F2b and Con-Ins F2c were derived from the species *C. floridulus*; Con-Ins Me1 was discovered in *C. memiae*; Con-Ins Im1 and Con-Ins Im2 were found in *C. imperialis*; Con-Ins Pa1, Con-Ins Ti1, Con-Ins Pu1, Con-Ins Vir and Con-Ins Bn1 were discovered in *C. planorbis*, *C. tribblei*, *C. pulicarius*, *C. virgo* and *C. bandanus*, respectively; Con-Ins Vr1 and Con-Ins Vr2 were found in *C. varius*; Con-Ins Ts1 and Con-Ins Ts2 were found in *C. tessulatus*; Con-Ins Eu1 and Con-Ins Eu2 were discovered in *C. eburneus*; Con-Ins Mr1 and Con-Ins Mr2 were discovered in *C. marmoreus*; Con-Ins Tx1 and Con-Ins Tx2 were discovered in *C.textile* and ILP was discovered in *C. victoriae* (Figure 3). The precursors of human insulin, zebrafish insulin, and conoinsulin are all composed of signal peptides and mature peptides [21,56]. The results of sequence comparison indicate that the signal peptide of insulin is highly conserved. Interestingly, we observed that unlike human insulin and zebrafish insulin, some conoinsulins possess a unique propeptide component, which might aid in protein folding during secretion, and prevent protein misfolding or aggregation [57,58]. 

Insulin, a natural hormone, has been proven to manifest an extensive assortment of aggregates with diverse structures and morphologies [51,59]. For instance, the mature region in insulin represents a functional domain encompassing both A, B and C chains, typically characterized by variability and interconnected through disulfide bonds, thereby facilitating the exertion of its pharmacological activity [56,60,61,62]. Like human insulin, the mature peptide of conoinsulin contains chains, albeit with notable differences. In insulin sequences, cysteine residues, which are largely conserved, predominantly cluster in the A and B chains. Typically, the A chain of insulin follows a CC-C-C cysteine pattern, while the B chain has a C-C pattern, leading to the formation of three disulfide bonds. In contrast, some conoinsulin A chains exhibit a C-CC-C-C pattern, while the B chain has a C-C-C pattern, resulting in four disulfide bonds. In addition, the human insulin B-chain contains a C-terminal segment, which plays an important role in the assembly of insulin dimers or hexamers [63,64,65,66]. Interestingly, some conoinsulins such as Con-Ins-G1, despite the noticeably short or partially absent C-terminus of the insulin B chain, still retain an affinity towards the human insulin receptor [52,67,68]. These findings imply that the C-terminal of the B-chain does not serve as a pivotal determinant influencing the binding affinity between insulin and the insulin receptor [67]. Moreover, the conoinsulin does not undergo dimerization or hexamerization, thereby allowing for rapid reaction kinetics surpassing those of current insulin medications. Cone snails produce conoinsulin not only for regulating blood sugar like most organisms, but also for predation purposes. This has led to the evolution of a wide variety of conoinsulins, akin to conotoxins, enhancing their efficacy on diverse prey. These attributes hold promising implications for the advancement of improved or novel therapeutic approaches for diabetes [52,68].

**Figure 3 toxins-16-00034-f003:**
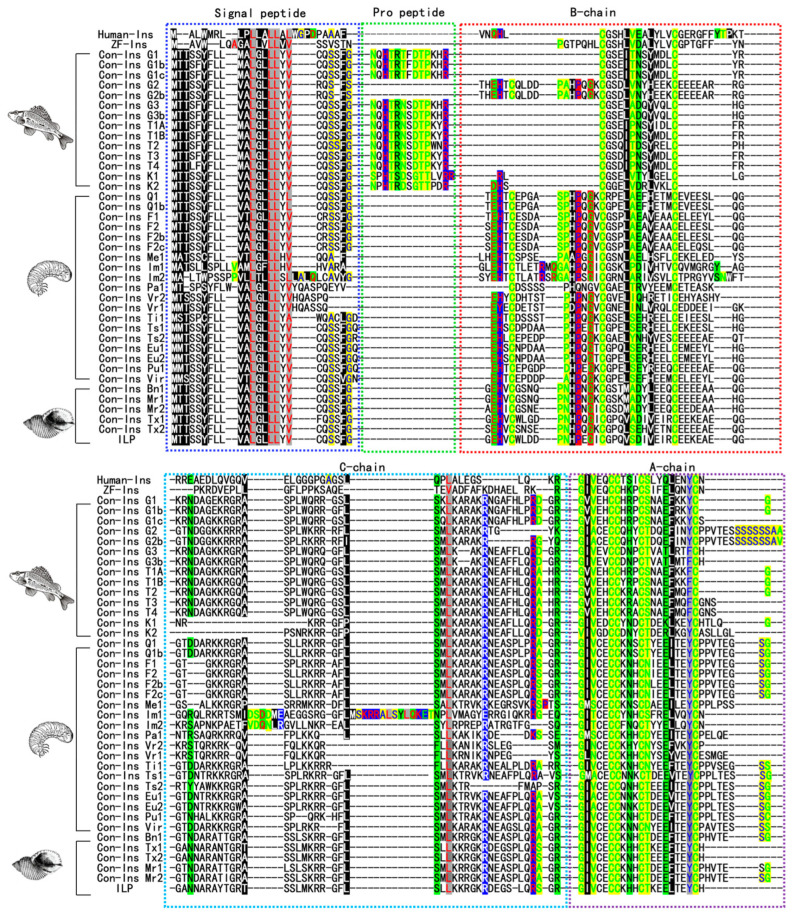
Alignment of insulin in humans, zebrafish, and different insulin sequences in the venom of various *Conus*. MEGA 7.0.14 software was used to create insulin sequence alignments and perform amino acid alignments on all peptide sequences, where MUSCLE algorithm was chosen to intelligently align amino acids [69,70]. Genedoc 2.7 software was used to export the sequence in FASTA format. B-chains and A-chains are enclosed within red and purple boxes, respectively. The conserved cysteine residues and small amino acids are highlighted in yellow. Aliphatic and aromatic amino acids are in red and blue fonts, respectively, on a grey background. Amphoteric and polar groups are in red and black fonts, respectively, on a green background. Negatively and positively charged amino acids are shown on a blue background in green and red. Proline and glycine amino acids are shown on a red background in blue and green. The hydrophobic amino acids are highlighted in black. ZF-Ins represents zebrafish insulin and ILP represents insulin-like peptide from *C. victoriae*.

The three-dimensional (3D) structure of Con-Ins G1 from *C. geographus* was solved using X-ray crystallography [71]. Con-Ins-G1 not only exhibits a high affinity for the human insulin receptor but also shares a striking similarity with the zebrafish insulin [41]. Biologically, Con-Ins G1 has the highest similarity to fish insulin [71], particularly in the A-chain, while the B-chain similarity is not as pronounced [41]. In addition, owing to its compact structure, Con-Ins G1 acts swiftly [72], which aligns well with the rapid predation strategy of cone snails. Using Con-Ins G1 (PDB 5JYQ) as a template, homologous modeling methods generated nine different conoinsulin variants corresponding to the dietary preferences of various cone snails, including G1b, G3, T1b, K1, F1, F2, Im1, Tx1, and Mr1. Meanwhile, 3D modeling of zebrafish insulin using human insulin (PDB 3I40) as a template revealed highly homologous structures (Figure 4) [63,73,74,75,76]. 

Through structural comparison, analysis reveals that all insulin molecules contain three α-helices, with two located in the A-chain and one in the B-chain. Additionally, they feature a hydrophobic core composed of non-polar residues, vital for their proper folding and structural integrity [77,78]. The conoinsulins from fish-hunting cone snails display a high degree of structural similarity with human insulin and zebrafish insulin. This resemblance likely correlates with their dietary characteristics, aiming to act on fish IRs to lower blood sugar levels for effective predation. However, conoinsulins from fish-hunting cone snails are significantly different from those of worm-hunting cone snails and mollusk-hunting cone snails, especially on the B-chain. The B-chain in the fish-hunting cone structure is shorter, which could be an adaptation for quicker prey capture.

**Figure 4 toxins-16-00034-f004:**
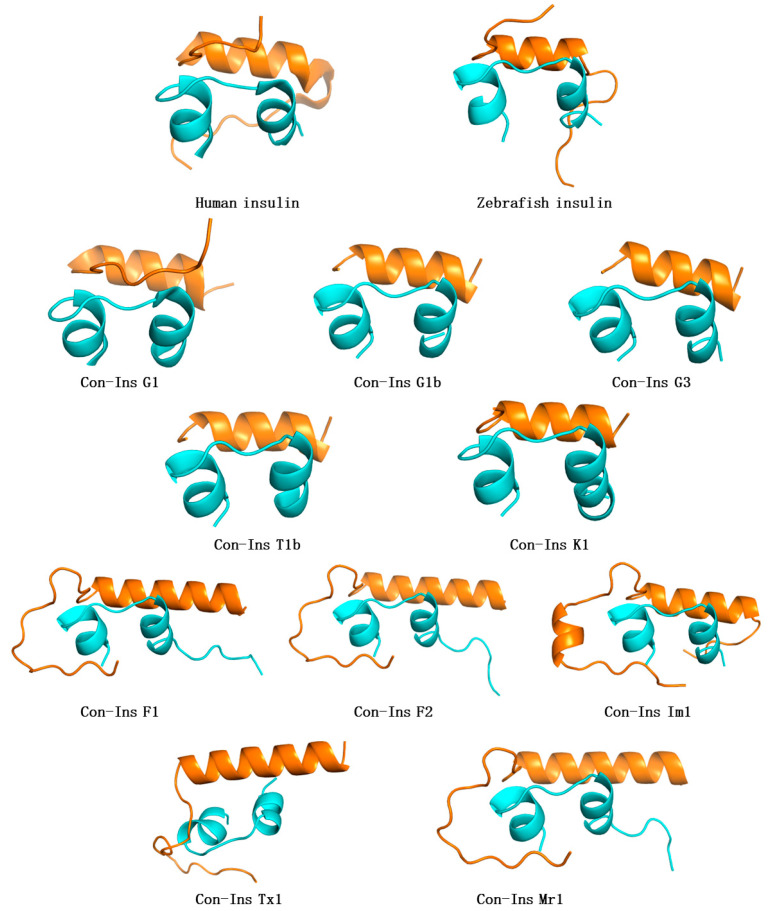
Comparison of the 3D structures of human insulin, zebrafish insulin, and conoinsulins. The cartoon representations of models of insulin variants are depicted, with A-chains and B-chains of each insulin in cyan and orange, respectively. The structures of human insulin (PDB 3I40) and Con-Ins G1(PDB 5JYQ) were sourced from the PDB database. Additional structures were obtained from AlphaFold Protein Structure Database (https://alphafold.ebi.ac.uk/). Protein 3D structure is predicted using homology computational structure prediction modeling from amino acid sequence [79]. SWISS-MODEL, available through the Expasy web server or Deep View software (Swiss Pdb-Viewer), is a fully automated server for the homology modeling of protein structures. The homologous sequences with highest sequence identity were assigned as templates, and then cartoon mode was used to build the model.

## 4. Evolutionary Relationship of Conoinsulin

To elucidate the evolutionary relationship of conoinsulins, we sourced conoinsulin sequences through a comprehensive search of the ConoServer and UniProt databases, followed by confirmation using the Blast algorithm. After removing duplicate sequences, a total of 38 different conoinsulin sequences and one insulin-like peptide were obtained, which were then analyzed using MEGA 7.0.14 software. Using the NJ method, we constructed a phylogenetic tree to facilitate the analysis of the evolutionary relationship among these conoinsulins (Figure 5). A diverse array of conoinsulins has been identified across *Conus* species: ranging from ancestral groups, such as worm-hunting *Conus* situated at the foundational levels of the food chain, to relatively advanced organisms, such as fish-hunting *Conus*.

Conoinsulin expression is ubiquitously observed in mollusk-hunting, worm-hunting, and fish-hunting *Conus* individuals. However, unlike mollusk-hunting *Conus* and worm-hunting *Conus*, only those fish-hunting *Conus* that use a net-hunting strategy express venom fish-like insulin [53]. Hence, among some fish-hunting *Conus* species, unique insulin variants serve as the principal components of their venomous arsenal, conferring a critical advantage in prey capture. 

Con-Ins Ti1, found in the worm-hunter *C. tribblei*, represents the most primitive venom insulins of cone snails. Subsequently, the phylogenetic tree diverged to include venom insulins produced by cone snails which specialize in feeding on fish and those that prey on snails. Among these fish-hunting *Conus*, the earliest appearing insulins are Con-Ins G2 and Con-Ins G2b originating from *C. geographus*. The venom insulins produced by cone snails with a fish-based diet, particularly the Con-Ins G1 originating from *C. geographus*, have attracted significant attention in recent research. It is noteworthy that the abundance of venom insulins from fish-hunting cone snails surpasses others, constituting approximately half of the phylogenetic tree encompassing *Conus* venom insulin. Following this, venom insulins produced by cone snails specializing in worm consumption display a moderate presence, whereas the quantity of venom insulins from mollusk-hunting cone snails remains relatively minimal.

According to the literature research, it has been found that all mollusk-hunting cone snails, most but not all worm-hunters and only a small subset of fish-hunters express conoinsulins [53]. Differing prey capture strategies may explain this difference. Some fish-hunting cone snails appear to release conoinsulins into the water to make an entire school of small fish hypoglycemic, thereby enhancing the cone snail’s ability to engulf multiple fish. In contrast, some fish-hunting species with no conoinsulins mainly capture fish by producing complex conotoxins that by cause hyper-excitability of the nervous system and rapid onset of a tetanic paralysis. There may be no role for a conoinsulin in this prey-capture strategy. Additionally, several worm-hunting species exhibit low or no levels of conoinsulin expression, indicating that conoinsulin may no longer be important in these species [51].

## 5. Conclusions and Perspectives

This review employed a comprehensive retrieval of cone snail insulin sequences from databases to evaluate the homology of venom insulin and analyze their sequence diversity. Additionally, a detailed phylogenetic tree was constructed to elucidate the evolutionary relationships among insulin sequences derived from cone snails with diverse dietary preferences. The insights gained provide a fundamental framework for future research into the targets and hypoglycemic activities of conoinsulins. 

Moreover, future advancements in sequencing technologies, particularly high-resolution tandem mass spectrometry, combined with venom gland transcriptome databases, hold great potential for enhancing our understanding. This integrated approach is expected to facilitate the discovery and identification of additional insulin sequences in cone snails. Such research will contribute significantly to our knowledge of the evolutionary patterns, diversity, and functional attributes of venom insulin in various subtypes of cone snails. Furthermore, such investigations open up promising avenues for the development and application of novel marine pharmaceuticals.

## Figures and Tables

**Figure 1 toxins-16-00034-f001:**
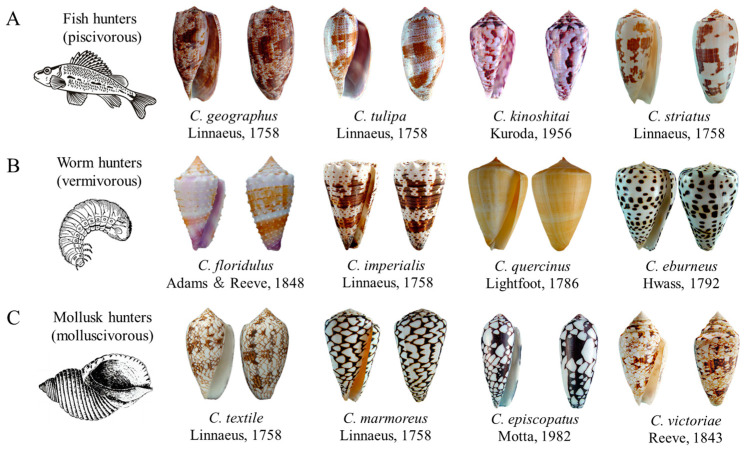
*Conus* species exhibit variances in their predator-prey relationships. (**A**) Fish hunters. (**B**) Worm hunters. (**C**) Mollusk hunters.

**Figure 2 toxins-16-00034-f002:**
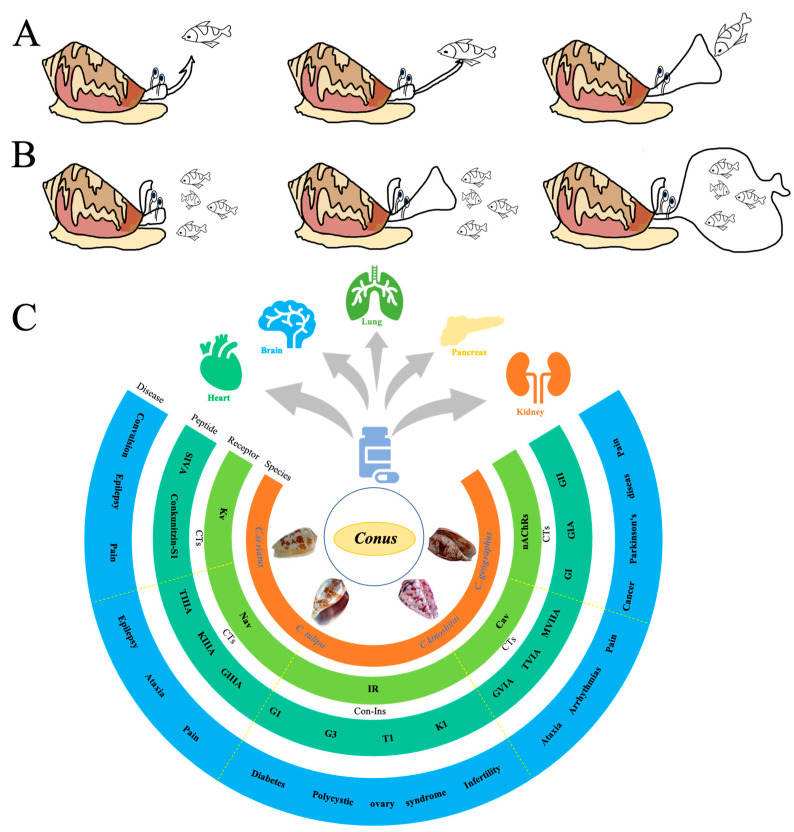
Roles as fishing weapons and medical value of conotoxins and conoinsulins. (**A**) Injection of the venom directly into the prey’s body to numb or poison it. (**B**) Release the venom into the surrounding water to numb the prey. (**C**) Diseases, receptors, and species related to different toxins and insulin. Representative cone snails are cited in the orange band. The light green band lists different targets for conotoxins or conoinsulins, including voltage-gated potassium ion channel (Kv), voltage-gated sodium channel (Nav), insulin receptor (IR), voltage-gated calcium channel (Cav), and nicotine acetylcholine receptor (nAChRs). The dark green band displays representative conotoxins or conoinsulins targeting different receptors. The blue band exemplifies typical diseases targeted by conotoxins or conoinsulins.

**Figure 5 toxins-16-00034-f005:**
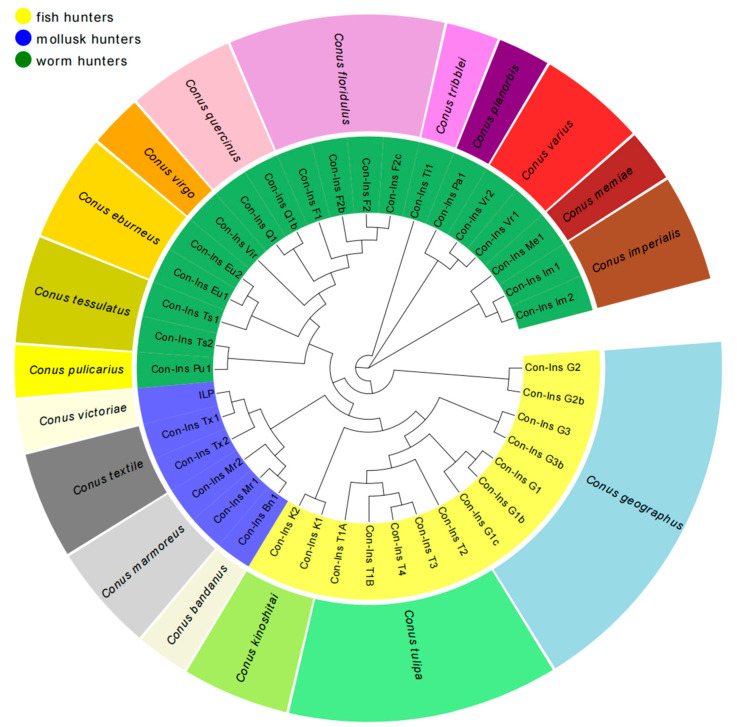
The phylogenetic tree of 38 different conoinsulin sequences and one insulin-like peptide from 18 species of Conus. The 39 peptide sequences were obtained from UniProt and ConoServer databases (www.uniprot.org/; http://conoserver.org/). The 39 peptide sequences were aligned using MEGA 7.0.14 software. A phylogenetic tree was established using a neighbor-Joining approach (bootstrap method 1000 and pairwise deletion 50%). The color in the inner circle of the figure indicates that insulin comes from three different dietary habits of Conus, while the color in the outer circle indicates different species of Conus.

## Data Availability

All insulin sequences are from UniProt and ConoServer databases (www.uniprot.org/; http://conoserver.org/).

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
