# Peer review of "Diversity and Evolutionary Analysis of Venom Insulin Derived from Cone Snails"

_toxins, 2024, doi:10.3390/toxins16010034_

Round 1

Reviewer 1 Report

Comments and Suggestions for Authors

I think this is a good review of the status quo of conus insulins. It is concese that it serves as a tool for searching for other possible insulins from other conus. I believe that this paper would be of interest to researchers in the field  However, I have some minor suggestion before acceptance:

-Latin names of species should always be in italic face (eg. line 28-conus, but there are more, please check carefully) and C. Geographus, (line 101)  letter g must be lowercase.

- Line 91 Conesnails - a space is missing

-Figure 2, section A, the image should be enhanced and credit given to the author of the original image.

Author Response

I think this is a good review of the status quo of conus insulins. It is concise that it serves as a tool for searching for other possible insulins from another conus. I believe that this paper would be of interest to researchers in the field However, I have some minor suggestion before acceptance:

-Latin names of species should always be in italic face (e.g., line 28-conus, but there are more, please check carefully) and C. Geographus, (line 101) letter g must be lowercase.

- Line 91 Conesnails - a space is missing

-Figure 2, section A, the image should be enhanced and credit given to the author of the original image.

ReplyWe greatly appreciate the reviewer’s positive feedback and commendation of our manuscript. We also believe that this article has the potential to engage readers deeply. Thank you very much for the valuable comments. We have diligently made revisions accordingly.

Through careful examination of the manuscript, we have ensured that all Latin names of species are consistently italicized (line 36 and so on), and the name of "C. geographus" has been correctly formatted properly (line 109) and the term "Conesnails" has been changed to "Cone snails" (line 99). For Figure 2, section A, we were unable to reach the author of the original image, so we have created our own illustrations depicting Conus predation, thereby circumventing any potential copyright issues.

Finally, thank you again and we look forward to receiving your satisfactory response.

Reviewer 2 Report

Comments and Suggestions for Authors

This manuscript entitled « Analysis of venom insulin diversity from cone snails as a weapon » (toxins-2706647) is intended to be a review on the conoinsulins, and to analyse the differences between the different isoforms that have been identified in different species. Although the purpose of this review is undoubtedly of particular interest, especially because of the enormous potential of conus toxins, in my opinion it falls far short of the objectives claimed by the authors.

More precisely, the authors claim that in their study « a comparative and analytical approach was employed to examine the sequence and structural attributes of insulin derived from cone snails » (Line 7-8).

In fact, the sequence analysis is limited to a few observations such as the presence or absence of a pro-peptide (line 243), the existence of several isoforms in certain species (line 238), but there is no commentary on these observations. What is the role of the pro-peptide? What distinguishes coninsulins with and without pro-peptide? What distinguishes the multiple variants found in a single species? And where is the comparative analysis of insulins in different cone snails?

The reader has to be satisfied with some irrelevant comments about the signal peptide (line 254 « the signal peptide region prominently showcases a concentration of hydrophobic amino acids, thereby highlighting its highly hydrophobic nature. » !) or the hydrophilicity of the C-peptide (line 259 « this region exhibits a marked attraction towards polar solvents. » !).

The structural analysis is concentrated in 4 lines (lines 190-194), where the reader learns that the A chain contains two helices, whereas the B chain contains a single one. I was surprised that the authors did a modelling of Con-Ins G1 when there is structural data for this peptide (pdb 5JQY).

In fact, there is no structural analysis but only a single, uninformative, illustration of the human insulin alongside that of zebrafish and C. geographus (Con-Ins G1) (Figure 3B)). Since Alphaflod allows to automatically obtain the structure from any sequence, it would have been interesting to compare, for example, Con-Ins G1 and G1b, to see where the difference(s) lie and for the authors to discuss these differences, or to compare the primitive insulin structure of worm-hunters with the evolved insulin structure of fish hunters. And, again, where is the comparative analysis of insulins in different cone snails?

At the end of the reading, the reader is very disappointed by the content of this review, which does not shed any particular light on this family of venom peptides.

Other comments

Line 7 and line 12 are redundant and abstract does not countain any conclusion.

Line 27 There are many references in this manuscript, and many of them seem inappropriate. For example, none of references 1-4 is a study of the geographical distribution of cone snails! On line 43, there are 7 references for a single observation, and at least one of them (ref 13 "Neuron and beta cell evolution: learning about neurons is learning about beta cells. ") seems completely irrelevant.

Line 56 exenatide should be used instead of exendin-4 tor avoid confusion.

Lines 51-64 and lines 66-77 The two paragraphs are similar and lead to the same conclusion. Exenatide is mentioned in both paragraphs, causing confusion.

Line 87 Although nice, figure 1 is irrelevant and should be removed

Lines 91-107 Useless talk about human poisoning. Should be removed

Lines 108-199 Useless talk about hunting strategies. Should be removed

Lines 121-140 Useless talk.

Line 164 « it is exciting that the venom produced by cone snails targets not only nAChRs ion channels » ??

Line 180-207 This chapter should be re-written. It starts by talking about insulin (lines 180-181), then conotoxins (lines 181-182) and then insulin again (lines 182-190). We are presented with Figure 3, with the modelling of Con-Ins G1 (without saying what is different between vertebrates and cone snails), then we come back to conoinsulin (line 195). In the end, it's all very confusing. Especially since the comparison of the different conotoxins in different species will be discussed in the next chapter, it is not known if what is described here is specific to Con-Ins G1.

Author Response

Dear reviewer,

We are deeply grateful for the reviewers and editors' insightful questions and comments regarding our article. Meanwhile, based on these questions and comments, we made careful revisions accordingly, and uploaded our point-by-point responses and revised manuscript (both clean and trackable versions).

Reviewer 2

This manuscript entitled « Analysis of venom insulin diversity from cone snails as a weapon » (toxins-2706647) is intended to be a review on the conoinsulins, and to analyze the differences between the different isoforms that have been identified in different species. Although the purpose of this review is undoubtedly of particular interest, especially because of the enormous potential of conus toxins, in my opinion it falls far short of the objectives claimed by the authors.

More precisely, the authors claim that in their study « a comparative and analytical approach was employed to examine the sequence and structural attributes of insulin derived from cone snails » (Line 7-8).

  1. In fact, the sequence analysis is limited to a few observations such as the presence or absence of a pro-peptide (line 243), the existence of several isoforms in certain species (line 238), but there is no commentary on these observations. What is the role of the pro-peptide? What distinguishes coninsulins with and without pro-peptide? What distinguishes the multiple variants found in a single species? And where is the comparative analysis of insulins in different cone snails?

The reader has to be satisfied with some irrelevant comments about the signal peptide (line 254 « the signal peptide region prominently showcases a concentration of hydrophobic amino acids, thereby highlighting its highly hydrophobic nature. » !) or the hydrophilicity of the C-peptide (line 259 « this region exhibits a marked attraction towards polar solvents. » !).

Reply: Thank you for the constructive feedback provided on our manuscript. We appreciate that this critical comment will help us improve the depth of this part of discussion on the conoinsulin sequences. Following the reviewer's suggestions, we have now enriched and rewritten the sequence analysis section titled “Diversity Analysis of Conoinsulin” starting from line 187.

Briefly, we discuss the composition of human insulin, zebrafish insulin, and conoinsulin, noting that each comprises signal peptides and mature peptides. Our analysis reveals that the signal peptide of insulin is highly conserved, serving primarily to direct the precursor peptides to the extracellular space, where the signal peptides are subsequently cleaved. Interestingly, we observed that unlike human insulin and zebrafish insulin, some conoinsulins possess a unique propeptide component, which aids in protein folding during secretion, and prevents protein misfolding or aggregation (lines 206-213).

Like human insulin, the mature peptide of conoinsulin contains A, B and C chains, albeit with notable differences. In insulin sequences, cysteine residues, which are largely conserved, predominantly cluster in the A and B chains. Typically, the A chain of insu-lin follows a CC-C-C cysteine pattern, while the B chain a C-C pattern, leading to the formation of three disulfide bonds. In contrast, some Conus insulin A chains exhibit a C-CC-C-C pattern, while the B chain a C-C-C pattern, resulting in four disulfide bonds (lines 218-224).

Lastly, we delve into the unique evolutionary adaptation of cone snails, which produce conoinsulin not for regulating blood sugar like most organisms, but for predation purposes. This has led to the development of a wide variety of conoinsulins, akin to conotoxins, enhancing their predatory efficacy on diverse prey (lines 232-235)."

2、The structural analysis is concentrated in 4 lines (lines 190-194), where the reader learns that the A chain contains two helices, whereas the B chain contains a single one. I was surprised that the authors did a modelling of Con-Ins G1 when there is structural data for this peptide (pdb 5JYQ). In fact, there is no structural analysis but only a single, uninformative, illustration of the human insulin alongside that of zebrafish and C. geographus (Con-Ins G1) (Figure 3B)). Since Alphaflod allows to automatically obtain the structure from any sequence, it would have been interesting to compare, for example, Con-Ins G1 and G1b, to see where the difference(s) lie and for the authors to discuss these differences, or to compare the primitive insulin structure of worm-hunters with the evolved insulin structure of fish hunters. And, again, where is the comparative analysis of insulins in different cone snails? At the end of the reading, the reader is very disappointed by the content of this review, which does not shed any particular light on this family of venom peptides.

Reply: We are immensely thankful for the reviewer's insightful comments, which significantly influenced the enhancement of our discussion on the 3D structures of conoinsulins. Following the reviewer's advice, we have thoroughly revised this part, including updates to the figures.

First of all, we accessed the structure of Con-Ins G1 (pdb 5JYQ) from the PDB database (we apologize for overlooking it in the first place). We then obtained 3D structures of various conoinsulins from AlphaFold. This led to a comprehensive comparison and analysis of the insulin structures of cone snails, considering both similar and different dietary habits. The revised section is presented as follows:

The 3D structure of Con-Ins G1 from C. geographus was solved using X-ray crystallography. Con-Ins-G1 not only exhibits a high affinity for the human insulin receptor but also shares a striking similarity with the zebrafish insulin. Biologically, Con-Ins G1 has the highest similarity to fish insulin, particularly in the A-chain, while the B-chain similarity is not as pronounced. In addition, owing to its compact structure, Con-Ins G1 acts swiftly, which aligns well with the rapid predation strategy of cone snails. Using Con-Ins G1(PDB 5JYQ) as a template, homologous modeling methods generated nine different conoinsulin variants corresponding to the dietary preferences of various cone snails, including G1b, G3, T1b, K1, F1, F2, Im1, Tx1, and Mr1. Meanwhile, 3D modeling of zebrafish insulin using human insulin (PDB 3i40) as a template revealed highly homologous structures (Figure 4).

Our structural comparison analysis reveals that all insulin molecules contain three α-helices, with two located in the A-chain and one in the B-chain. Additionally, they feature a hydrophobic core composed of non-polar residues, vital for their proper folding and structural integrity. The conoinsulins from fish-hunting cone snails display a high degree of structural similarity with human insulin and zebrafish insulin. This resemblance likely correlates with their dietary characteristics, aiming to act on fish insulin receptors to lower blood sugar levels for effective predation. However, conoinsulins from fish-hunting cone snails are significantly different from worm-hunting cone snails and mollusk-hunting cone snails, especially on the B-chain. The B-chain in the fish-hunting cone structure is shorter, which could be an adaptation for quicker prey capture (lines 251-271).

Please refer to the manuscript for detailed images.

Figure 4. Comparison of the 3D structures of human insulin, zebrafish insulin, and conoinsulins. The cartoon representations of models of insulin variants are depicted, with A-chains and B-chains of each insulin in cyan and orange, respectively. The structures of human insulin (PDB 3i40) and Con-Ins G1(PDB 5JYQ) were sourced from the PDB database. Additional structures were obtained from AlphaFold Protein Structure Database (https://alphafold.ebi.ac.uk/).

Other comments

  • Line 7 and line 12 are redundant and abstract does not countain any conclusion.

Reply: We are grateful to the reviewer for the guidance in refining the abstract. To provide a more comprehensive overview, we have enhanced the abstract with definitive findings: “The comparative analysis reveals that there are significant variations in the sequences and three-dimensional structures of venom insulins from cone snails with different feeding habits. Notably, the venom insulin of piscivorous cone snails exhibits a greater similarity to humans and zebrafish insulins. It is important to emphasize that these venom insulins play a crucial role in the predatory strategies of these cone snails.” (lines 16-20).

(2) Line 27 There are many references in this manuscript, and many of them seem inappropriate. For example, none of references 1-4 is a study of the geographical distribution of cone snails! On line 43, there are 7 references for a single observation, and at least one of them (ref 13 "Neuron and beta cell evolution: learning about neurons is learning about beta cells. ") seems completely irrelevant.

Reply: Thank you very much for the careful review of the references. We apologize for not carefully checking the references. The inappropriate references have been removed and new references have been added as below:

  1. Bingmiao, G.; Chao, P.; Jiaan, Y.; Yunhai, Y.; Junqing, Z.; Qiong, S. Cone Snails: A Big Store of Conotoxins for Novel Drug Discovery. Toxins 2017, 9, e0196982.
  2. Lewis, R.J. Ion Channel Toxins and Therapeutics: From Cone Snail Venoms to Ciguatera. Therapeutic Drug Monitoring 2000, 22, 61-64.
  3. Favreau, P.; Gall, F.L.; Benoit, E.; Molgó, J. A review on conotoxins targeting ion channels and acetylcholine receptors of the vertebrate neuromuscular junction. Acta physiologica, pharmacologica et therapeutica latinoamericana: órgano de la Asociación Latinoamericana de Ciencias Fisiológicas y [de] la Asociación Latinoamericana de Farmacología 1999, 49, 257-267.
  4. Robinson, S.D.; Li, Q.; Bandyopadhyay, P.K.; Gajewiak, J.; Yandell, M.; Papenfuss, A.T.; Purcell, A.W.; Norton, R.S.; Safavi-Hemami, H. Hormone-like peptides in the venoms of marine cone snails. Gen Comp Endocrinol 2017, 244, 11-18, doi:10.1016/j.ygcen.2015.07.012.
  5. Norton, R.S.; Olivera, B.M. Conotoxins down under. Toxicon: An International Journal Devoted to the Exchange of Knowledge on the Poisons Derived from Animals, Plants and Microorganisms 2006, 48.

(3) Line 56 exenatide should be used instead of exendin-4 to avoid confusion.

Reply: We thank the reviewer for carefully checking our manuscript. The error has been corrected (line 69).

(4) Lines 51-64 and lines 66-77 The two paragraphs are similar and lead to the same conclusion. Exenatide is mentioned in both paragraphs, causing confusion.

Reply: We agree with the reviewer's comments. The overlapped content in these two paragraphs indeed causes confusion. Therefore, we have deleted the discussion about exenatide and the conclusion from the first paragraph and revised the second paragraph as below. “Exenatide, found in the saliva of the Hilan lizard, is one of the most classical examples of venom toxin derivatives. It effectively regulates glycemic levels by stimulating glucose-dependent insulin secretion, and has been developed for the treatment of diabetes” (lines 69-72).

(5) Line 87 Although nice, figure 1 is irrelevant and should be removed 

Reply: We appreciate the reviewer’s suggestion and recognize the merit in the reviewer’s perspective. However, we believe that for readers less acquainted with cone snails, this figure offers a clear and engaging depiction of the species mentioned in the manuscript, enhancing their understanding of cone snails with different dietary habits. Therefore, we kindly request permission to retain Figure 1, considering its educational value to our audience.

(6) Lines 91-107 Useless talk about human poisoning. Should be removed. Lines 108-199 Useless talk about hunting strategies. Should be removed. Lines 121-140 Useless talk.

Reply: We are thankful for the reviewer’s input. We consider the discussed contents important for readers unfamiliar with the complete predation process of cone snails. It offers valuable insights into the evolutionary development of conoinsulin as a predatory tool. Therefore, we wish to maintain these sections and kindly seek the reviewer’s approval for the same.

(7) Line 164 « it is exciting that the venom produced by cone snails targets not only nAChRs ion channels » ??

Reply: We thank the reviewer for highlighting the ambiguity in our initial wording. To enhance clarity and coherence, we have made revisions to this section as below: “The comprehensive literature review reveals that the crude venom of cone snails targets not only voltage-gated ion channels, such as Nav, Kv and Cav channels, but also acts on nAChR and insulin receptors (IRs) [Figure 2C]” (lines 170-172).

(8) Line 180-207 This chapter should be re-written. It starts by talking about insulin (lines 180-181), then conotoxins (lines 181-182) and then insulin again (lines 182-190). We are presented with Figure 3, with the modelling of Con-Ins G1 (without saying what is different between vertebrates and cone snails), then we come back to conoinsulin (line 195). In the end, it's all very confusing. Especially since the comparison of the different conotoxins in different species will be discussed in the next chapter, it is not known if what is described here is specific to Con-Ins G1.

Reply: We sincerely thank the reviewer for this valuable comment. Accordingly, we have revised the manuscript to eliminate the redundant content and consolidate useful information in a logical manner. For more details, please refer to the updated section "3. Diversity Analysis of Conoinsulin" in the manuscript (lines 187-283).

Reviewer 3 Report

Comments and Suggestions for Authors

In this manuscript, the authors present a short review on the diversity of cone snail insulin, with the aim to gain insights into the phylogenetic correlations among insulin variants from various cone snail species.  To this, they performed a comparative analysis to examine the sequence and structural attributes of insulin derived from  cone snails. A comprehensive set of cone snail insulin sequences was retrieved from databases to assess the homology of venom insulin and to analyse their sequence diversity. Then, a phylogenetic tree was constructed to trace the lineage of insulin sequences from cone snails with varied diets, providing insight into their evolutionary interconnections.

Overall, this manuscript is well written, and enjoyable to read. Although it seems very didactic, as it introduces a lot of basic knowledge on the biology of conotoxins, it is still very interesting for non-experts in the field, as it lists many commercial drugs derived from conotoxins.

I am not sure whether this manuscript presents any new points that have not already been previously discussed in other articles, as by doing just a short Pubmed search, I can find articles more or less similar to this one (see doi: 10.3390/biomedicines8080235, doi: 10.1093/molbev/msw174, doi: 10.1073/pnas.1423857112, doi: 10.7554/eLife.41574, for example).

In conclusion, I think it will be of interest to the large audience of Toxins, and therefore I recommend its publication.

Minor comments:

The title is somewhat misleading, as it does not correctly introduce the content of the manuscript. In particular, the word “weapon” is not necessary.

Second, the aims of the manuscript are not fully addressed. In fact, as a key contribution of the manuscript, the authors wrote: “Investigating the homology and evolutionary lineage of diverse conoinsulin variants, along with examining the resemblance between conoinsulin and human insulin, provides a solid groundwork for comprehending the diversity and pharmacological characteristics of conoinsulins”.  I have to say that, from the work presented here, it would be arduous to understand  the pharmacological characteristics of these toxins.

Author Response

Dear reviewer,

We are deeply grateful for the reviewers and editors' insightful questions and comments regarding our article. Meanwhile, based on these questions and comments, we made careful revisions accordingly, and uploaded our point-by-point responses and revised manuscript (both clean and trackable versions).

In this manuscript, the authors present a short review on the diversity of cone snail insulin, with the aim to gain insights into the phylogenetic correlations among insulin variants from various cone snail species. To this, they performed a comparative analysis to examine the sequence and structural attributes of insulin derived from cone snails. A comprehensive set of cone snail insulin sequences was retrieved from databases to assess the homology of venom insulin and to analyse their sequence diversity. Then, a phylogenetic tree was constructed to trace the lineage of insulin sequences from cone snails with varied diets, providing insight into their evolutionary interconnections.

Overall, this manuscript is well written, and enjoyable to read. Although it seems very didactic, as it introduces a lot of basic knowledge on the biology of conotoxins, it is still very interesting for non-experts in the field, as it lists many commercial drugs derived from conotoxins.

I am not sure whether this manuscript presents any new points that have not already been previously discussed in other articles, as by doing just a short Pubmed search, I can find articles more or less similar to this one (see doi: 10.3390/biomedicines8080235, doi: 10.1093/molbev/msw174, doi: 10.1073/pnas.1423857112, doi: 10.7554/eLife.41574, for example).

In conclusion, I think it will be of interest to the large audience of Toxins, and therefore I recommend its publication.

Reply:We appreciate the reviewer’s insightful feedback. In the manuscript, we have  referenced the papers suggested by the reviewer. Although some of our discussions draw upon ideas from these sources, this manuscript uniquely integrates and examines all the available insulin sequences from cone snails in the current databases. We compare and analyze these sequences, and construct an evolutionary tree of conoinsulin based on the varying dietary habits of cone snails. This comprehensive approach distinguishes our work from other articles and presents a key highlight of this manuscript.

Minor comments:

1、The title is somewhat misleading, as it does not correctly introduce the content of the manuscript. In particular, the word “weapon” is not necessary.

Reply:We thank the reviewer for this valuable suggestion. Following this advice, we have made modifications to the title by removing "as a Weapon" from the title. The new title is "Diversity and Evolutionary Analysis of Venom Insulin Derived from Cone Snails".

2、Second, the aims of the manuscript are not fully addressed. In fact, as a key contribution of the manuscript, the authors wrote: “Investigating the homology and evolutionary lineage of diverse conoinsulin variants, along with examining the resemblance between conoinsulin and human insulin, provides a solid groundwork for comprehending the diversity and pharmacological characteristics of conoinsulins”. I have to say that, from the work presented here, it would be arduous to understand the pharmacological characteristics of these toxins.

Reply:We are thankful to the reviewer for pointing this out. Following this suggestion, we have revisited the manuscript and provided a more accurate explanation of its purpose. “Key Contribution: Conoinsulin serves not only as a natural, inherent predatory weapon but also as a valuable resource in marine medicine research. By exploring the homology and evolutionary lineage of diverse conoinsulin variants, along with examining the resemblance between conoinsulin and human insulin, we provide a theoretical framework to comprehend the diversity of venom insulin used in the predation strategies of cone snails with different feeding habits. This enhanced understanding paves the way for further scientific exploration in this domain” (lines 24-29).

Thanks for all the help. We are looking forward to your reply at your earliest convenience.

Round 2

Reviewer 2 Report

Comments and Suggestions for Authors

I appreciate the changes that have been made to the original version of the manuscript and thank the authors for taking my comments into account.

I find that the manuscript is now more balanced and easier to read. I no longer have any objection to its publication in toxins.

Author Response

We greatly appreciate the reviewer’s positive feedback and commendation of our manuscript. We also believe that this article has the potential to engage readers deeply. 

Finally, thanks for your help again.